# Laboratory Monitoring of Direct Oral Anticoagulants (DOACs)

**DOI:** 10.3390/biomedicines9050445

**Published:** 2021-04-21

**Authors:** Claire Dunois

**Affiliations:** HYPHEN BioMed, Sysmex Group, 95000 Neuville sur Oise, France; cdunois@hyphen-biomed.com

**Keywords:** DOAC, monitoring, screening assays, quantitative assays

## Abstract

The introduction of direct oral anticoagulants (DOACs), such as dabigatran, rivaroxaban, apixaban, edoxaban, and betrixaban, provides safe and effective alternative to previous anticoagulant therapies. DOACs directly, selectively, and reversibly inhibit factors IIa or Xa. The coagulation effect follows the plasma concentration–time profile of the respective anticoagulant. The short half-life of a DOAC constrains the daily oral intake. Because DOACs have predictable pharmacokinetic and pharmacodynamic responses at a fixed dose, they do not require monitoring. However in specific clinical situations and for particular patient populations, testing may be helpful for patient management. The effect of DOACs on the screening coagulation assays such as prothrombin time (PT), activated partial thromboplastin time (APTT), and thrombin time (TT) is directly linked to reagent composition, and clotting time can be different from reagent to reagent, depending on the DOAC’s reagent sensitivity. Liquid chromatography–mass spectrometry (LC-MS/MS) is considered the gold standard method for DOAC measurement, but it is time consuming and requires expensive equipment. The general consensus for the assessment of a DOAC is clotting or chromogenic assays using specific standard calibrators and controls. This review provides a short summary of DOAC properties and an update on laboratory methods for measuring DOACs.

## 1. Introduction

Direct oral anticoagulants (DOACs) constitute first-line therapy used for many thromboembolic indications, such as prevention and treatment of venous thromboembolism (VTE) and stroke prevention in atrial fibrillation (AF) [1,2]. This class of medication consists of the direct thrombin (FIIa) inhibitor (dabigatran) or direct factor Xa (FXa) inhibitors (apixaban, rivaroxaban, edoxaban, and betrixaban). The advantages of DOACs compared with vitamin K antagonists (VKAs) are a more favorable bleeding profile, limited food, and drug interactions, and they do not require routine monitoring, except under some specific conditions [3,4,5,6,7]. DOAC testing may be useful in the case of critical clinical situations, including drug accumulation in long-term treatment, overdosage, thrombotic or bleeding events, acute stroke, trauma, forthcoming surgery, or emergency surgery [8,9,10,11]. Monitoring can be useful in patients with obesity (≥120 kg) or low body weight (≤50 kg) when the drug concentration is, respectively, lower or higher than expected. These results confirm the requirement of plasma level measurement for management of certain patients [12,13].

After 10 years of DOAC use, the question of DOAC monitoring in key situations where clinical uncertainty exists is still being debated. Monitoring can be done using routine coagulation tests or specific tests. This review provides a tool for the test monitoring choice.

## 2. DOAC Pharmacokinetics and Pharmacodynamics

### 2.1. Dabigatran

Dabigatran etexilate (Pradaxa) is the prodrug of dabigatran that directly, specifically, and reversibly inhibits free and clot-bound thrombin by interaction with the thrombin-active site. Because of its high polarity, dabigatran is not bioavailable after oral administration. Consequently, a prodrug of dabigatran, dabigatran etexilate, was developed to improve gastrointestinal absorption [14]. The absolute bioavailability of the prodrug is about 3% to 7%, and dabigatran presents a plasma protein binding of about 35%, implying that dabigatran has no displacement interactions without pharmacokinetic and pharmacodynamic degradation (Table 1). After oral intake, dabigatran etexilate is rapidly hydrolyzed to the active form by nonspecific and ubiquitous carboxylester hydrolases. This enzymatic process produces two intermediate metabolites: the metabolite BIBR 1087 and the metabolite BIBR 951. Then, the acylglucuronide of the carboxylate functional group is formed, and this is the major metabolite in humans. Because the proteolytic reactions converting dabigatran etexilate to dabigatran do not imply cytochrome P450 enzymes (CYP) or other oxidoreductases, the risk of drug–drug interactions is low [15]. The peak plasma concentrations (C_max_) of dabigatran are measured in healthy volunteers within 1.5 to 3 h after oral administration. C_max_ is about 175 ng/mg for a dose of 150 mg, and the concentration 12 h after dosing is about 30% to 50% of C_max_ (Table 2). Dabigatran has a prolonged elimination phase, with a mean plasma terminal half-life of 12 to 14 h, regardless of the dose [16,17]. Unchanged dabigatran is predominantly eliminated by renal clearance (80%) [15]. Dabigatran etexilate, but not dabigatran, is a probe for the P-glycoprotein (P-gp) transporter.

### 2.2. Rivaroxaban

Rivaroxaban (Xarelto) is a direct selective FXa inhibitor that specifically and reversibly inhibits both prothrombinase and clot-bound FXa [22]. Rivaroxaban absorption is almost complete with oral bioavailability of about 80% to 100% at a dose of 10 mg. Peak maximum plasma concentrations (C_max_) are achieved within 2 to 4 h after intake (Table 1), with a C_max_ of about 250 ng/mL for a dose of 20 mg (Table 2) [23]. Extremes of body weight (<50 kg or >120 kg) may slightly affect the plasma concentration of rivaroxaban (less than 25%). In humans, rivaroxaban exhibits reversible, high plasma protein binding of approximately 92–95%. After oral intake, the predominant form circulated is unchanged rivaroxaban and no major active metabolites are found in plasma. Rivaroxaban has a dual pathway of elimination: urinary excretion of unchanged drug (renal clearance of 40%) and elimination by the hepatobiliary route [24]. Approximately 36% of the parent drug is processed in two major metabolites (metabolites M11 and M12). Rivaroxaban is metabolized by cytochrome P450 (CYP) enzymes CYP3A4/3A5 and CYP2J2 and hydrolyzed [25,26]. Rivaroxaban does not induce or inhibit of any major CYP isoforms even if the metabolism of rivaroxaban is mediated by this enzymatic pathway. Because the renal clearance of rivaroxaban involves active transporters P-gp and breast cancer resistance protein (Bcrp (ABCG2)), strong inhibitors of these transporters may reduce the renal clearance [27]. Rivaroxaban has a smaller elimination phase, with a plasma terminal half-life of 5 to 9 h in healthy subjects (Table 1) [23,28,29].

### 2.3. Apixaban

Apixaban (Eliquis) is a potent and direct inhibitor of FXa that specifically and reversibly inhibits both prothrombinase and clot-bound FXa [28]. Food has no significant effect on the oral bioavailability of apixaban, which is approximately 60% at a dose of 2.5 mg and approximatively 50% at oral doses up to 10 mg [29,30]. After oral administration, the peak maximum plasma concentrations (C_max_) occur within 3 to 4 h, with a C_max_ of about 170 ng/mL at a dose of 5 mg (Table 1 and Table 2). Extremes of body weight (<50 kg or >120 kg) may affect the plasma concentration with an approximate increase of 27% for low body weight and with an approximate decrease of 31% for high body weight [31]. In human plasma, apixaban is mainly bound to albumin and presents an approximative protein binding of 87% [32]. After oral intake, the major form of apixaban found in human plasma is the unchanged drug and no active circulating metabolites are present. The elimination of apixaban engages several pathways, including renal clearance of the unchanged parent drug (27%), hepatobiliary excretion (2.5%), and direct fecal elimination (56%) [33]. Apixaban is metabolized by O-demethylation, hydroxylation, and sulfation of hydroxylated O-demethyl apixaban to form metabolite M1 (25%), metabolite M2, and metabolite M14. Hepatic metabolism occurring via the cytochrome P450 (CYP) enzyme CYP3A4/5 principally, with a minor contribution of enzymes CYP1A2, CYP2C8, CYP2C9, CYP2C19, and CYP2J2 [34]. Apixaban is a substrate for transporters P-gp and Bcrp (ABCG2), and as for rivaroxaban, strong inhibitors of these transporters may reduce renal clearance. Co-administration of strong dual inhibitors P-gp and CYP3A4 with apixaban is generally not recommended, but the effect is moderate to minor when only one of these inhibitors is administered [35]. Apixaban has an intermediate elimination phase, with a plasma terminal half-life of 10 to 14 h in healthy subjects (Table 1) [28,36].

### 2.4. Edoxaban

Edoxaban (Lixiana or Savaysa) is a highly selective direct inhibitor of FXa that reversibly inhibits both prothrombinase and clot-bound FXa. Following oral administration as edoxaban tosylate, the drug is rapidly absorbed. Edoxaban has an oral bioavailability of about 62% at a dose of 60 mg. The peak maximum plasma concentrations (C_max_) are achieved within 1 to 2 h, with a C_max_ of about 170 ng/mL at a dose of 60 mg (Table 1 and Table 2) [37,38]. Extremes of body weight do not seem to affect the plasma concentration of edoxaban regardless of the dosing regimens. Plasma protein binding of edoxaban is approximately 55% for the parent drug and 80% for metabolite M4. After oral intake, edoxaban is transformed to many metabolites by carboxylesterase-1 (CES-1) to form metabolite M4 (10% of the parent drug); by hepatic metabolism of the cytochrome P450 (CYP) enzyme CYP3A4/5 mediating the formation of metabolites M5, M6, and M8 (less than 4% of the parent drug); by hydrolysis (M1); and by glucuronidation to form an *N*-glucuronide metabolite M3 [39]. The metabolites M4 (active) and M1 are the major metabolites; other metabolites, M2, M3, and M5, are only in trace amounts. The only active metabolites are M4, M6, and M8, representing less than 10% of the total anticoagulant activity. Elimination of edoxaban follows different pathways: urinary excretion of the unchanged parent drug (50%), hepatobiliary secretion, and metabolism [40]. Each of the metabolites M1, M4, and M6 contributes to less than 2% of renal clearance, and each of the metabolites M4, M6, and M8 represents less than 2% of fecal excretion [39]. Even if edoxaban is a substrate for the transporter P-gp, the inhibitor of this transporter results in a slight increase in the drug plasma concentration. Co-administration of an inhibitor of P-gp and a strong inhibitor of CYP results in a moderate effect [41,42]. Edoxaban has an intermediate elimination phase, with a plasma terminal half-life of 9 to 11 h in healthy subjects (Table 1) [43].

### 2.5. Betrixaban

Betrixaban (Bevyxxa or Dexxience) is a direct and selective inhibitor of FXa that reversibly inhibits prothrombinase and clot-bound FXa. This anticoagulant shows a high affinity for FXa and a strong specificity compared with thrombin, FVIIa or FIXa [44]. Betrixaban has an oral bioavailability of about 34% at a dose of 80 mg. After oral administration, the peak maximum plasma concentrations (C_max_) occur within 3 to 4 h, with a C_max_ of about 46 ng/mL at a dose of 80 mg (Table 1 and Table 2). high-fat food and low-fat food reduce the C_max_ by about 50% and 70%, respectively. In vitro, plasma protein-binding is approximatively 60%. Contrary to other DOACs, renal excretion of the parent drug and inactive metabolite is low, contributing only to 6% to 13% of the total clearance, and elimination involves little or no metabolism by cytochrome P450 (CYP) enzymes (<1%). Betrixaban is predominantly eliminated unchanged by the hepatobiliary route (>82%) via P-gP efflux pumps. Because betrixaban has no CYP interactions reported, the risk of drug interaction with the inhibitors of CYP is minor [45]. Inversely, co-administration of strong inhibitors of P-gp with betrixaban lead to elevated betrixaban plasma concentrations, and these drugs affecting P-gp have to be used with caution in patients under betrixaban therapy [45]. Compared to other DOACs, betrixaban has an extended elimination phase, with a terminal half-life of 35 to 45 h, giving a low peak-to-trough concentration, leading to a global anticoagulant effect extended Check whether edits retain the intended meaning. 24 h.

## 3. DOAC Inter-Patient Variability

### 3.1. Management of Patients under DOAC Therapy

Direct oral anticoagulants are easy-to-use oral drugs that offer simple dosing with a fixed dose and short half-lives, but this strategy of DOAC management may be not optimal for some patient populations. Consequently, the plasma concentration of DOACs can vary from patient to patient. The inter-patient variability is reported to be approximately 30% for dabigatran and apixaban and 30–40% for rivaroxaban [46]. This variability is particularly observed in the case of hepatic and renal insufficiency, extreme weight, and co-medication, with a strong effect on the metabolic pathway [47,48,49]. In patients who have an increased bleeding risk, dose adjustments are recommended. For patients under dabigatran, rivaroxaban, or edoxaban therapy for stroke prevention in AF, dose reduction is based on the estimated creatinine clearance (Table 3). For patients under apixaban therapy for stroke prevention in AF, two of three of the following criteria are required for dose reduction: body weight ≤ 60 kg, age ≥ 80 years, and serum creatinine ≥ 1.5 mg/dL (Table 3) [50]. For acute VTE, for recurrent VTE prevention, or for post-operative VTE prophylaxis, DOAC dose reduction is mainly based on the estimated creatinine clearance (Table 3).

To avoid any bleeding events, the renal function of patients under DOAC therapy should be monitored at least once per year, especially if renal dysfunction is suspected (acute myocardial infarction, acute decompensated heart failure, diabetes, use of diuretics, certain co-medications, dehydration, hypovolemia). In the case of relevant impairment, a dose reduction or discontinuation of the drug is required.

Another way to decide on a dose reduction is to monitor the DOAC plasma concentrations in patients with increased risk of bleeding: patients aged >75 years, history of bleeding, and dual therapy with drugs affecting hemostasis, such as nonsteroidal anti-inflammatory drugs (NSAIDs), acetylsalicylic acid, and platelet aggregation inhibitors Indeed, DOAC plasma concentrations are well correlated with bleeding events in patients under anticoagulant therapy [51,52,53,54].

### 3.2. Management of Patients under Multi-Drug Therapy

Drug interactions can also be criteria for dose adjustment, especially with strong inhibitors or inducers of CYP3A4 and P-glycoprotein (P-gp) that may increase (inhibitors) or decrease (inducers) DOAC plasma concentrations to a clinically relevant degree, leading to increase bleeding risk (inhibitors) or thrombotic risk (inducers) (Table 4) [42]. A combination of dabigatran with strong P-gp inhibitors such as ketoconazole, glecaprevir, or pibrentasvir is not recommended, and a combination with P-gp inhibitors such as cyclosporine, itraconazole, nelfinavir, posaconazole, ritonavir, saquinavir, tacrolimus, or tipranavir should be managed with caution; concomitant medication with P-gp inducers such as the anticonvulsants carbamazepine, phenytoin, and St. John’s Wort is contraindicated. Rivaroxaban and apixaban are not recommended in patients receiving concomitant treatment with strong inhibitors of both CYP3A4 and P-gp, such as azole-antimycotics (e.g., ketoconazole, itraconazole, voriconazole, and posaconazole) or HIV protease inhibitors (e.g., ritonavir), and a combination with strong inducers of CYP3A4/P-gp (e.g., rifampicin, phenytoin, carbamazepine, phenobarbital, or St. John’s Wort) should be avoided [27,35]. A combination of edoxaban with P-gp inhibitors such as cyclosporine, dronedarone, erythromycin, or ketoconazole requires a dose reduction to 30 mg once daily, and concomitant treatment with rifampicin has to be avoided [41].

### 3.3. Pre-Operative Management of Patients under DOAC Therapy

A patient under DOAC therapy who require an invasive procedure or surgical intervention has to stop the treatment before the intervention. The management of these patients is based on the half-life of the drug and its persistence in blood. The clearance of a DOAC is strongly influenced by renal/hepatic function, and before a high-bleeding-risk procedure, the duration of DOAC discontinuation can differ from patient to patient according to creatinine clearance, age, comorbid disease, bleeding history, and concomitant treatment [8,9,10,11]. The current strategy for elective surgery with standard-to-low bleeding risk is to stop dabigatran 1 to 2 days before intervention if creatinine clearance (CrCL) ≥ 50 mL/min and 2 to 3 days before intervention if CrCl ≤ 50 mL/min. For elective surgery with a high bleeding risk, the strategy is to stop dabigatran 2 to 3 days before intervention if creatinine clearance (CrCL) ≥ 50 mL/min and 4 days before intervention if CrCl ≤ 50 mL/min. For direct anti-Xa, the duration of discontinuation for elective surgery with standard-to-low bleeding risk is at least 24 h before intervention and for elective surgery with a high bleeding risk, the duration is at least 2 days. In patients requiring an urgent intervention associated with a high bleeding risk (if the procedure cannot be delayed) or in the case of life-threatening or uncontrolled bleeding and when a DOAC concentration above 50 ng/mL is found by the laboratory (overdosage or short time after intake), a DOAC reversal agent is recommended. Idarucizumab (Praxbind) is the reversal agent for dabigatran, and Andexanet alfa (Andexxa-Ondexxya) is the reversal agent for anti-Xa. Idarucizumab is administered intravenously at a dose of 5 g. Andexanet alfa is administered as an intravenous bolus of about 400 mg if the timing of the last dose of apixaban ≥ 8 h or about 800 mg if the timing of the last dose of apixaban < 8 h or a dose of >10 mg rivaroxaban or 5 mg apixaban. [55,56]. In any case, before elective surgery or urgent intervention or an invasive procedure, the concentration of the DOAC has to be lower than 30 ng/mL to avoid any bleeding risk. To improve the management of these patients, measurement of the DOAC plasma concentration seems to be a good option.

## 4. DOAC Laboratory Testing

The optimal laboratory assay to monitor DOAC concentrations depends on test availability, the level of information required (drug presence (qualitative test) or drug concentration (quantitative test)), and the turnaround time for the result.

### 4.1. Liquid Chromatography–Mass Spectrometry (LC-MS/MS)

LC-MS/MS is a sophisticated method to measure plasma concentrations of all DOACs by detecting the drug molecules in a pretreated plasma sample. LC-MS/MS directly quantifies drug concentrations and does not require calibration using a known concentration of the DOAC [57]. This methodology has a high degree of specificity and sensitivity, with a limit of detection (LoD) and a limit of quantitation (LoQ) between 0.025 and 3 ng/mL, depending on the drug and technology. The analytical ranges of measurement are between 5 and 500 ng/mL for all DOACs, which is sufficient to quantify concentrations at peak and at trough in most patients (Table 2). For DOACs with active metabolites, an additional measurement must be done to quantify each metabolite. For example, the prodrug dabigatran etexilate is metabolized to dabigatran. Approximately 20% of dabigatran is conjugated by glucuronosyltransferases to the pharmacologically active glucuronide conjugates; thus two active forms are found in plasma: free and conjugated to glucuronide. Dabigatran glucuronide adds approximately 20% of anticoagulant activity. To optimize LC-MS/MS quantitation (one-time measurement), the sample must undergo alkaline hydrolysis to split the conjugate and allow measurement of the total dabigatran [46,58]. For edoxaban, because of the difficulty to measure all active metabolites, only edoxaban and metabolite M4 are measured. However, by comparison with other quantitative assays, a bias can be observed, especially from 10 to 24 h after intake (Figure 1 and Figure 2) [19]. LC-MS/MS is considered the gold standard method for DOAC measurement [19]. However, it requires expensive equipment and well-trained people and has limited throughput. Due to its complexity, only few highly specialized laboratories have this technology and can perform DOAC measurement. Additional limitations are the absence of standardization or harmonization of LC-MS/MS assays and the lack of an international standard for calibration [59]. LC-MS/MS is extensively used for research and clinical trials, but it cannot be used in routine conditions for rapid testing in laboratories [60,61].

### 4.2. Routine Coagulation Screening Assays

The effect of DOACs on screening coagulation assays such as prothrombin time (PT), activated partial thromboplastin time (APTT), and thrombin time (TT) has been widely studied [62]. These methods have a high throughput and are routinely available in all clinical laboratories, conferring an important benefit in the case of emergency situations to support clinical decisions. However, variability of the result can be important from reagent to reagent, depending on the sensitivity of the reagent to the DOAC. Screening tests can be used as first-line tests, but they are insufficient to assess the degree of anticoagulant effects, as seen with the International Normalized Ratio (INR) for management of VKA therapy.

#### 4.2.1. Direct Thrombin Inhibitors (DTIs)

The dabigatran dose–response curve with APTT reagents is curvilinear, flattening at approximatively 200 ng/mL [58]. The clotting time at a given concentration varies between reagents, possibly related to differences in reagent composition. With a required dabigatran concentration of approximatively 400 ng/mL, prolongation time is twofold with APTT [63,64]. Despite these limitations, the APTT can be a useful screening assay and could provide an approximative estimate of dabigatran levels at therapeutic concentrations (Table 5). Depending on reagent sensitivity, the result can be semi-quantitative or only qualitative, and for some APTTs, the presence of dabigatran may not be reliably detected.

The effect of dabigatran on the clotting time with a PT test is lesser. The clotting time can be prolonged with regard to the dabigatran plasma concentration, but it is not systematically prolonged at peak maximum plasma concentrations (C_max_) and never at trough concentrations (Table 5). Currently, at the C_max_ of dabigatran, the International Normalized Ratio (INR) is less than 1.5 [62]. In this way, PT does not seem to be a suitable assay to measure or detect dabigatran [65].

Because of its formulation, the TT is highly sensitive to dabigatran, and a normal clotting time is highly related to no dabigatran or low concentrations in the plasma sample. For concentrations lower than or close to 30 ng/mL, the clotting time is significantly prolonged and the time for clotting can be 10-fold prolonged or in many cases above the limit of measurement at C_max_ [19]. The effect of dabigatran on the assay is directly dependent on the concentration of thrombin in TT reagents. Accordingly, a prolonged TT cannot be automatically associated with a high dabigatran level, and the TT is not suited for accurate quantitation, but it can be used to exclude the presence of dabigatran (Table 5) [65].

#### 4.2.2. Direct Factor Xa Inhibitors (DiXaIs)

The APTT dose–response curve is non-linear, and this non-linearity strongly increases with concentrations of DOAC anti-Xa [62]. The APTT is slightly prolonged at the C_max_ of anti-FXa drugs, with comparable anticoagulant effects of betrixaban and edoxaban, followed by rivaroxaban and apixaban, which have lesser effects [66]. In a large number of apixaban-treated patients, concentrations of up to 200 ng/mL are not detected with the APTT test. Compared to other DiXaIs, rivaroxaban has a higher sensitivity to the APTT [67]. However, sensitivity, which depends on the reagent used, is generally insufficient to monitor treatment at therapeutical concentrations, especially at lower concentrations [64]. Because of that, the APTT may not reliably detect the presence of DiXaIs (Table 5).

The first studies performed in the early stage of clinical development of direct anti-FXa inhibitors showed that the rivaroxaban concentration could be measured with the PT to help with the clinical management of certain treated patients. Rivaroxaban prolongs the PT with a concentration-dependent time for clotting [62]. The PT is more sensitive than the APTT to rivaroxaban and edoxaban, but depending on the reagent used, wide variability is observed among reagents. In most patients under rivaroxaban therapy, the PT is prolonged at C_max_ and C_trough._ In patients under edoxaban therapy, the clotting time at C_max_ is prolonged with most PT reagents (Table 5). However, the magnitudes of change and PT prolongation are reagent dependent. The anticoagulant effects of edoxaban seems stronger, followed by betrixaban and rivaroxaban with comparable anticoagulant effects, slightly weaker than edoxaban. Apixaban produced slight or no anticoagulant effects [66]. By consequence, apixaban concentrations of up to 200 ng/mL have no effect on the PT. For all anti-Xa DOACs, sensitivity is insufficient at concentrations lower than or close to 30 ng/mL [19]. By the way, the PT can be useful to exclude excess amounts at the trough of rivaroxaban or edoxaban in plasma but only using sensitive reagents. In contrast, the PT cannot be used to measure concentrations of apixaban [66,67,68,69,70,71]. The PT does not seem to be an ideal candidate for the follow-up of plasma concentrations in DiXaIs for routine.

The TT is not affected by DiXaIs (Table 5).

### 4.3. Quantitative Routine Assays

#### 4.3.1. Direct Thrombin Inhibitors (DTIs)

Current methods for quantitative measurement of dabigatran include the ecarin clotting time (ECT), chromogenic ecarin assay (ECA), chromogenic anti-FIIa assay, and diluted TT. All these methods must be calibrated with the appropriate standard to be able to quantify dabigatran.

The ECT has a linear dose–response relationship, and the clotting time is directly linked to dabigatran concentrations. The ECT can be used for measuring dabigatran concentrations over the therapeutic range. Formulations of the ECT and ECA are similar except for signal detection; the first one is based on the time taken to form a clot, and the second one relies on color generation. These assays present important variations from lot to lot due to major component concentrations. Fibrinogen and prothrombin deficiencies can impact the trueness of the test. The ECT and ECA show a good correlation with LC-MS/MS when calibrated with a commercial standard [19,59,60,61,62,63].

The diluted TT is a clotting assay based on thrombin time in which patient plasma is diluted. First, plasma is diluted in a buffer, then it is mixed with a normal pool plasma, and coagulation is triggered by adding purified human thrombin essentially in α-thrombin form. Dilution overrides the significant sensitivity to dabigatran noted with thrombin time. The measurement ranges span all the concentrations expected, including C_max_ or cut-off concentrations. The diluted TT has a linear dose–response relationship with clotting times inversely proportional to dabigatran concentrations. Commercial assays report LoD ranges of 2–8 ng/mL and LoQ ranges of 20–30 ng/mL. These limits (LoD and LoQ) can be improved using a low-range protocol (lower sample dilution) in combination with low calibrators and controls (HEMOCLOT™ TI low-range protocol with an LoQ of about 5 ng/mL) [72,73,74].

Few commercial kits are available for quantitative determination of dabigatran using the chromogenic anti-FIIa method. Thrombin is incubated with neat or diluted patient plasma, and residual thrombin hydrolyzes the thrombin-specific chromogenic substrate, releasing paranitroaniline (pNA). The amount of pNa released, measured by the optical density generated per minute, is inversely proportional to the concentration of dabigatran in the sample. Commercial assays report an LoD of approximately 15 ng/mL. This assay is also available for a low concentration using a lower sample dilution with a similar LoD and LoQ than clotting anti-FIIa assays [70]. Some kits may contain a heparin-neutralizing agent that can be used in the case of post-operative heparin-to-dabigatran bridging therapy. An excellent correlation and correspondence of concentrations measured is observed with the clotting (HEMOCLOT™ TI) or the chromogenic (BIOPHEN™ DTI) anti-FIIa assays and with the LC-MS/MS method (Figure 3) [74].

#### 4.3.2. Direct factor Xa Inhibitors (DiXaIs)

Chromogenic anti-FXa assays have been used in clinical laboratories for several decades for monitoring heparin and heparin-like therapies. These anti-FXa kinetic methods have been adapted from existing chromogenic assays for the measurement of DiXaIs, with drug-dedicated calibrators and controls. These different chromogenic assays have different dynamic ranges and tested plasma dilutions to allow good linearity and parallelism [75,76]. These one-stage chromogenic assays are based on the inhibition of a constant amount and in excess of FXa by DiXaI. The residual FXa hydrolyzes a specific chromogenic substrate, releasing pNA. The amount of pNa released, measured by the optical density generated per minute, is inversely proportional to the concentration of DiXaI in the sample. Analytical and clinical performance studies of these DiXaI-specific anti-FXa chromogenic assays demonstrate good accuracy and specificity. The measurement ranges of DiXaIs cover the expected levels after therapeutic doses, but sensitivity of standard assays do not allow one to correctly assess a concentration lower than or about 30 ng/mL due to a limited LoQ. A procedure using lower sample dilution in combination with low calibrators and controls (BIOPHEN™ Heparin low-range protocol (LRT)) may be used to improve lower limits. With an LoD and LoQ less than 5 and 10 ng/mL, respectively, this assay is allowed to measure low plasma concentrations of anti-Xa drugs [76].

A specific two-stage assay is also available for DiXaI testing (BIOPHEN™ DiXaI). The principle of this two-stage method is the inhibition of a constant and limited amount of human FXa by DiXaI in the presence of a high sodium chloride concentration. The residual factor Xa hydrolyzes the FXa-specific chromogenic substrate, releasing pNA, and the absorbance measured is inversely proportional to the DiXaI concentration. Under these ionic conditions, heparins can no longer interact with AT, and the assay is then specific for DiXaIs. Furthermore, in the case of bridging therapy, the results provided by the kinetic method can be influenced by the presence of heparinoids. In addition, the key advantage of this chromogenic two-stage assay is its insensitivity to the presence of antithrombin-dependent FXa inhibitors such as heparins and derivatives [75]. An excellent correlation is observed with the kinetic or two-stage anti-FXa assays (BIOPHEN™ Heparin LRT; BIOPHEN™ DiXaI) and with the LC-MS/MS method. These anti-FXa assays measure the global anti-Xa activity of edoxaban and all its active metabolites (M4-M6-M8), and they can be used for assessing low and high concentrations, when LC-MS:MS measures only edoxaban or edoxaban + metabolite M4 (Figure 1 and Figure 4).

## 5. Conclusions

One advantage of DOAC use over VKA is the predictable pharmacokinetic and pharmacodynamic profile, which mitigates the need for frequent monitoring to guide dosing. Over the past decade, DOACs have been investigated in many clinical scenarios. However, clinical studies are conducted with a fixed dose of DOACs and do not evaluate clinical outcomes based on drug concentrations or coagulation assays. No evidence-based recommendation for drug concentration measurements, coagulation tests, assay standardization, or target therapeutic ranges has been clearly established for DOACs [21]. DOACs have varying effects on coagulation assays, and the clinical utility of these tests is still being determined. In addition, thresholds for coagulation tests are not established, because they differ considerably in their diagnostic performances. Each laboratory should know the sensitivity of its own PT and APTT tests to dabigatran, rivaroxaban, apixaban, edoxaban, and betrixaban and advise on interpretation. The general consensus for the assessment of DOACs is clotting or chromogenic assays using specific standard calibrators and controls.

The measurement of residual DOAC concentrations in emergency situations can help clinical decision making and optimal patient management. The decrease in DOAC activity following withdrawal can be slow, and the effect measured at 24 h after the last intake remains high in some patients (≥50 ng/mL). The currently accepted threshold for DOACs is <30 ng/mL, but current quantitative assays, especially anti-FXa assays, have an LoQ at the threshold or over, making them unreliable for assessing concentrations below the LoQ [77]. The ability to measure residual DOAC levels using low range methodologies is particularly important in these emergent situations to assess DOAC concentrations and to guide clinical decisions.

## Figures and Tables

**Figure 1 biomedicines-09-00445-f001:**
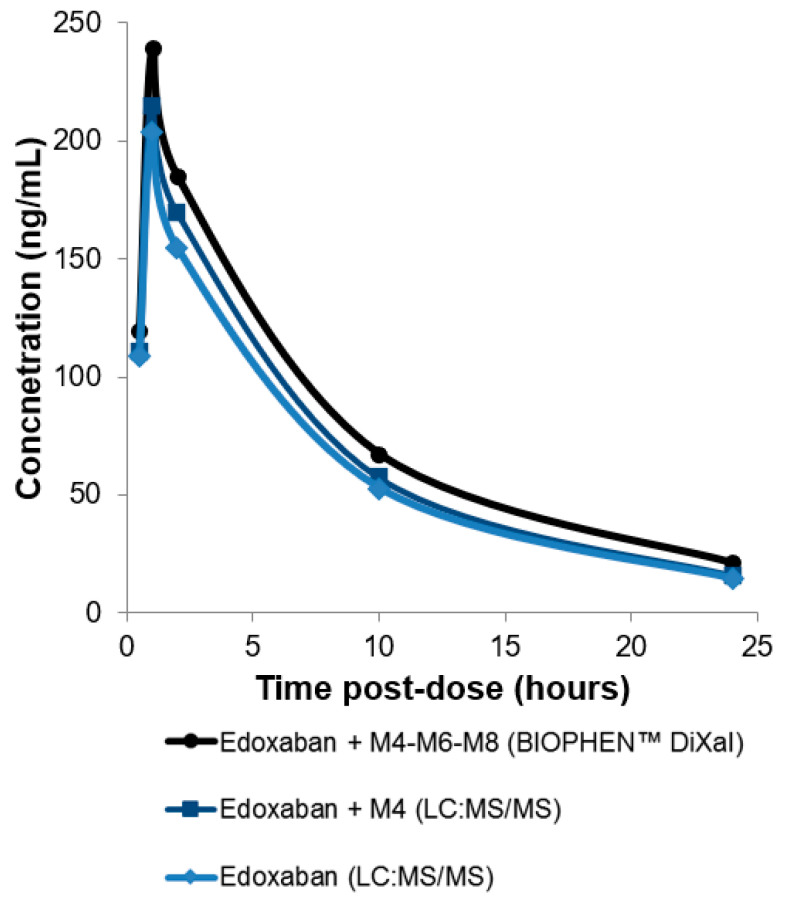
Measurement of edoxaban and active metabolites with a bioassay (BIOPHEN™ DiXaI) and edoxaban or edoxaban + metabolite M4 measurement with liquid chromatography–mass spectrometry (LC-MS/MS) in plasma samples from healthy volunteers (collected at different times after intake). Following edoxaban administration, rapid absorption occurred, resulting in peak plasma concentrations at 1 to 2 h, followed by a decline phase (data not published).

**Figure 2 biomedicines-09-00445-f002:**
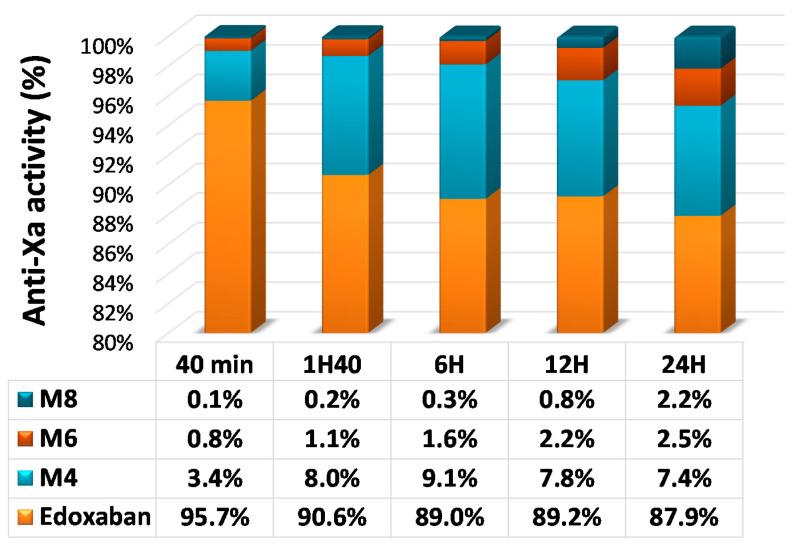
Plasma concentrations of edoxaban and active metabolites after a single oral dose of 60 mg of [^14^C] edoxaban. Adapted from Bathala et al. [45].

**Figure 3 biomedicines-09-00445-f003:**
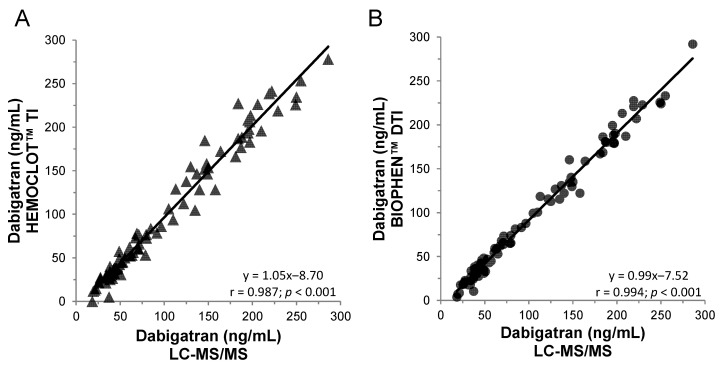
Correlation between anti-FIIa bioassays and the LC-MS/MS method on 100 dabigatran plasma samples from the RELY or phase 1 studies. (**A**) HEMOCLOT™ TI and (**B**) BIOPHEN™ direct thrombin inhibitor (DTI).

**Figure 4 biomedicines-09-00445-f004:**
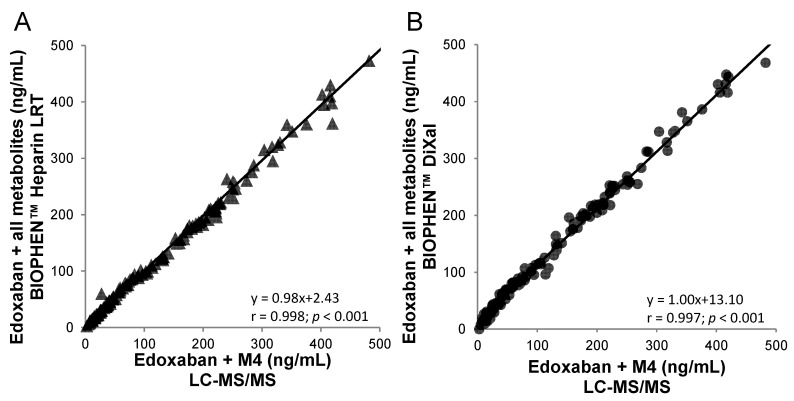
Correlation between anti-FXa bioassays and the LC-MS/MS (edoxaban + M4) method on 144 edoxaban plasma samples from healthy volunteers. (**A**) BIOPHEN™ Heparin LRT and (**B**) BIOPHEN™ direct factor Xa inhibitor (DiXaI) (data not published).

**Table 1 biomedicines-09-00445-t001:** Comparative pharmacology of direct oral anticoagulants (DOACs).

	Dabigatran	Rivaroxaban	Apixaban	Edoxaban	Betrixaban
Action	Anti-IIa	Anti-Xa	Anti-Xa	Anti-Xa	Anti-Xa
Frequency of intake	Twice daily	Once daily	Twice daily	Once daily	Once daily
T_max_ (h)	1.5–3	2–4	3–4	1–2	3–4
T_trough_ (h)	12	24	12	24	24
Half-life (h)	12–14	5–9	10–14	9–11	35–45
Bioavailability	3–7%	80–100%	50%	62%	34%
Protein binding	35%	92–95%	87%	40–59%	60%
Renal clearance (%)	80%	40%	27%	50%	6–13%

T_max_: time at peak plasma drug concentration; T_trough_: time at trough plasma drug concentration.

**Table 2 biomedicines-09-00445-t002:** DOAC plasma concentrations.

	Dabigatran ^a^	Rivaroxaban ^b^	Apixaban ^c,d^		Edoxaban ^e^	Betrixaban ^f^
Frequency of intake	Twice daily	Once daily	Twice daily	Twice daily	Once daily	Once daily
Dose	150 mg	20 mg	5 mg	10 mg	60 mg	80 mg
C_max_ (ng/mL)	175 (117–275)	249 (184–343)	171 (91–321)	251 (111–572)	170 (125–245)	46 (5–117)
C_trough_ (ng/mL)	91 (61–143)	44 (12–137)	103 (41–230)	120 (41–335)	36 (19–62)	17 (16–22)

^a^ Mean (25/75th percentile) in stroke prevention. ^b^ Mean (5/95th percentile) in stroke prevention. ^c^ Median (5/95th percentile) in stroke prevention. ^d^ Median (5/95th percentile) in venous thromboembolism (VTE) treatment [18]. ^e^ Median (1.5 interquartile range) in stroke prevention [19]. ^f^ Median C_max_ in healthy subjects [20] and median (25/75th percentile) C_trough_ in the population with VTE risk [21]. C_max_: peak plasma drug concentration; C_trough_: trough plasma drug concentration.

**Table 3 biomedicines-09-00445-t003:** Management of DOAC in atrial fibrillation (AF) and venous thromboembolism patients.

	Dabigatran	Rivaroxaban	Apixaban	Edoxaban
AF dosage	150 mg twice daily	20 mg once daily	5 mg twice daily	60 mg once daily
Criteria for adjustment	CrCL 30–50 mL/min No recommendation: CrCL < 30 mL/min or HD	CrCL 15–50 mL/min No recommendation: CrCL < 15 mL/min or HD	2 of 3 required: body weight ≤ 60 kg age ≥ 80 years SCr ≥ 1.5 mg/dL	CrCL 15–50 mL/min Avoid use if: CrCL < 15 mL/min CrCL > 95 mL/min No recommendation for HD
Dose adjustment	110 mg twice daily	15 mg once daily	2.5 mg twice daily	30 mg once daily
VTE treatment Recurrent VTE VTE prophylaxis	150 mg twice daily 150 mg twice daily 150 mg twice daily	15 mg (21 days) then 20 mg once daily 10 or 20 mg once daily 10 mg once daily	10 mg (7 days) then 5 mg twice daily 2.5 mg twice daily 2.5 mg twice daily	60 mg once daily
Criteria for adjustment	CrCL 30–50 mL/min No recommendation: CrCL < 30 mL/min	Avoid use if: CrCL < 30 mL/min or HD	No recommendation: CrCL < 25 mL/min	CrCL: 15–50 mL/min or body weight ≤ 60 kg Avoid use if: CrCL < 15 mL/min
Dose adjustment	110 mg twice daily			30 mg once daily

AF: atrial fibrillation; VTE: venous thromboembolism; CrCL: creatinine clearance; HD: hemodialysis; SCr: serum creatinine.

**Table 4 biomedicines-09-00445-t004:** Drug interactions.

	Dabigatran	Rivaroxaban	Apixaban	Edoxaban	Betrixaban
Drug interaction (inhibitors)	P-gp	Strong CYP3A4+P-gp	Strong CYP3A4+P-gp	P-gp	P-gp
Plasma concentration	↑	↑	↑	↑	↑
Dose adjustment	Reduce or avoid	Avoid	Reduce or avoid	Reduce (VTE)	Avoid: CrCL < 30 mL/min
Drug interaction (inducers)	P-gp	Strong CYP3A4 or P-gp	Strong CYP3A4 or P-gp	P-gp	P-gp
Plasma concentration	↓	↓	↓	↓	↓
Dose adjustment	Avoid	Avoid	Avoid	Avoid with Rifampin	No recommendation

VTE: venous thromboembolism; CrCL: creatinine clearance; P-gp, P-glycoprotein.

**Table 5 biomedicines-09-00445-t005:** Changes in coagulation screening tests caused by DOACs.

	Dabigatran	Rivaroxaban	Apixaban	Edoxaban	Betrixaban
PT/INR ^a,b^	↑ at C_max_	↑	Slight or no change	↑	↑
APTT ^a,b^	↑	↑at C_max_	Slight or no change	↑ at C_max_	↑ at C_max_
TT	↑at C_through_	No change	No change	No change	No change

^a^ Reagent dependent. ^b^ Concentration dependent. APTT: activated partial thromboplastin time; PT: prothrombin time; TT: thrombin time.

## Data Availability

Not applicable.

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
