# Peer review of "Laboratory Monitoring of Direct Oral Anticoagulants (DOACs)"

_biomedicines, 2021, doi:10.3390/biomedicines9050445_

Round 1

Reviewer 1 Report

In the present review, the author provides a summary of DOAC properties and an update of laboratory methods for measuring DOAC.

  • The review is well written and complete in presentation. However, the content is quite generic and similar to information available from previously published articles.
  • A further concern could be represented by conflict of interest considering that the only author listed works for HYPHEN BioMed a French company with products for anticoagulant monitoring widely presented in the review.
  • DOACs have different indications (AF, VTE …) but the Author only reported treatment schedules in AF patients. Table 2 should be updated also reporting other schedules currently available.
  • A further table with criteria for dose adjustment in specific settings could be useful
  • All paragraphs up to the end of page 14 are related to pharmacokinetic and pharmacodynamic properties of DOACs. Although interesting in the frame of a comprehensive review, this can make the review less focused on laboratory monitoring issue.

Minor points:

  • Some typo errors are present
  • Considering that DOACs are used since 10 years in clinical practice it is better to avoid to define this a “new” class of medication

Author Response

Dear Reviewer,

Thank you for your helpful remarks and your relevant comments which I have taken into account to improve this review.

Concerning the potential conflic of interrest, I worked during more than 15 years for public institute in haemostasis reasearch. Writting this review, my objective was not to promote reagents from one or other supplier but to give an overview of the possibilities in term of DOAC measurement with the benefit and the inconveniance of each type of assay.

Originally, table 2 was an illustration of approximative peak concentration find in plasma, to answer your request , I added the information at 10 mg of apixaban.

Concerning , dose adjustment, I added a table and a paragraph to explain circumstances of DOAC therapy adjustment.

With the new part on DOAC inter-patient variability, I hope the message of review will be more comprehensive.

Thank you for your time.

Best regards,

Dr C. DUNOIS

Reviewer 2 Report

Laboratory Monitoring of Direct Oral Anticoagulants (DOAC)

The review is well structured and focused on the current scientific knowledge about pharmacokinetics, pharmacodynamics and laboratory monitoring of direct oral anticoagulants.

Major remarks

Introduction

I think that dosing instructions for DOAC should not be written in the introduction section but in an independent brief paragraph. This way the review would be more complete.

Minor remarks

The non-valvular adjective for atrial fibrillation should be deleted since the indication to DOACs remains despite the presence of several valvulopathies.

Some reference should be revised.

For example:

Line 382: the page number is available for reference 1.

Some typos are present throughout the text so English should be revised.

For example:

1) the sentence below (line 10) is not clear and should be rewritten.

“Because DOAC have predictable pharmacokinetic and pharmacodynamic responses at fixed dose and they do not required monitoring”.

2) line 20 the word proper-ties is perhaps properties.

3) line 74: concertation is perhaps concentration.

4) missing a bracket in Table 2

5) line 167 …. the risk of drug interaction with inhibitors of CYP during is minor. The sentence should be rewritten.

6) line 169 … have to be used with caution 169 fin patient. I think in patient

7) line 183 between 0.025 183 ng/mL at 3 ng/mL

8) line 346 Although this chromogenic 346 two-stage assay is insensitive to the presence of antithrombin-dependent FXa inhibitors 347 such as heparins and derivatives [71].  This sentence should be rewritten.

Author Response

Dear Reviewer,

Thank you for your helpful remarks and your relevant comments which i have taken into account to improve this review.

As suggested, I moved the dosing instruction from introduction to a new paragraph, I also added tables to explained conditions of dose adjustment and drug interactions.

With this new part on DOAC inter-patient variability, I hope the message of review will be more comprehensive.

I also correct minor points except the page number for reference 1 which are not available on pubmed and using Zotero, the only information I found is the volume (7) and the number (3).

Thank you for your time.

Best regards,

Dr C. DUNOIS

Reviewer 3 Report

Direct oral anticoagulants (DOAC) are first line therapy used for many thromboembolic indications, such as prevention and treatment of venous thromboembolism and stroke prevention in non-valvular atrial fibrillation. DOAC testing may be useful in case of critical clinical situations, including drug accumulation in long term treatment, overdosage, thrombotic or bleeding events, acute stroke, trauma, forthcoming surgery or for emergency surgery, especially with the availability of DOAC reversal agents. After 10 years of DOAC use, the question of DOAC monitoring in key situations where clinical uncertainty exists, is still being debated. Monitoring can be done using routine coagulation tests or specific tests. Therefore, this review provides a tool for test monitoring choice.

The manuscript is written on fifteen pages, four of which provide references. From my point of view it provides the basic characteristics typical for a complex review type of the article.

The article could be published after major revision according to the comments to the authors.

Content suggestions:

  1. The authors provide standard review that seems quite concise. As the authors concluded that “DOAC testing may be useful in case of critical clinical situations, including drug accumulation in long term treatment, overdosage, thrombotic or bleeding events, acute stroke, trauma, forthcoming surgery or for emergency surgery, especially with the availability of DOAC reversal agents”, I would like to ask them to add more details about monitoring of hemostasis and subsequent modification of DOAC dose in these situations.
  2. For similar reasons, I would like to ask the authors to add more data about drug interactions as they may be the reason why to more intensively monitor hemostasis in patients using DOACs.
  3. Could the authors include also information about adverse events in the patients using these drugs and their contraindications ?

The article could be published after major revision according to the comments to the authors.

Author Response

Dear Reviewer,

Thank you for your helpful remarks and your relevant comments which i have taken into account to improve this review.

As suggested, I added a new paragraph on dose adjustment with a table to clarify circumstances of DOAC therapy adjustment. I also added a paragraph and a table to detailed drug interactions and consequences on DOAC plasma concentrations.

With this new part on DOAC inter-patient variability, I hope the message of review will be more comprehensive.

Thank you for your time.

Best regards,

Dr C. DUNOIS

Round 2

Reviewer 2 Report

The manuscript has been improved.

Minor remark:

in Table 3: dabigatran is not recommended for CrCl <30 ml/min and dose adjustment for CrCl between 30 and 50 ml/min could be as high as 110 mg twice a day since a dose of 75 mg is not available in all countries.

Author Response

Dear Reviewer,

Thank you for this second review and pertinent new comment.

You are right, information on the table was provided by europeen agency, I adjust it to be on line with FDA Dabigatran monograph.

Thank you again.

Best regards,

Dr Claire DUNOIS

Reviewer 3 Report

The presented manuscript has been mostly corrected in response to the suggestions. After the revision, the provided data and interpretation of the results became more clear.

However, despite the initial recommendation, there are only limited notes about additional laboratory tests used for the specific monitoring of haemostasis in the association with DOACs (the article is about monitoring, so I would like to kindly ask the authors for the addition of the information about  the tests, such as dRVVT, dTT/DTI, ECT/ECA, PICT...).

Moreover, I would like to repeatedly ask the authors to add mentioned more details about reversal agents – their dosing, when exactly to use it, how...).

I sincerely thank the authors for resubmitting the manuscript and explaining the obscure points from the previous version.

The article could be published after minor corrections as suggested above.

Author Response

Dear reviewer,

Thank you for this second review and new comments.

I understand you would like to have information on interfernces of DOAC on routine tests but it was not the aim of this review. I already discussed interferences on routine test and used of DOAC stop in previous posters and article. It is for that reason that I focussed this rview only on DOAC monitoring.
Concerning your second comment, I added complementary information on reversal agents. I hope it will be more informative now.

Best regards,

Dr Claire DUNOIS